# Clipped histone H3 is integrated into nucleosomes of DNA replication genes in the human malaria parasite *Plasmodium falciparum*

Abril Marcela Herrera-Solorio[1], Shruthi Sridhar Vembar[2,3,4,†], Cameron Ross MacPherson[2,3,4], Daniela Lozano-Amado[1], Gabriela Romero Meza[1], Beatriz Xoconostle-Cazares[5], Rafael Miyazawa Martins[2,3,4], Patty Chen[2,3,4], Miguel Vargas[1], Artur Scherf[2,3,4,*] (iD) & Rosaura Hernández-Rivas[1,**] (iD)

## Abstract

Post-translational modifications of histone H3 N-terminal tails are key epigenetic regulators of virulence gene expression and sexual commitment in the human malaria parasite *Plasmodium falciparum*. Here, we identify proteolytic clipping of the N-terminal tail of nucleosome-associated histone H3 at amino acid position 21 as a new chromatin modification. A cathepsin C-like proteolytic clipping activity is observed in nuclear parasite extracts. Notably, an ectopically expressed version of clipped histone H3, PfH3p-HA, is targeted to the nucleus and integrates into mononucleosomes. Furthermore, chromatin immunoprecipitation and next-generation sequencing analysis identified PfH3p-HA as being highly enriched in the upstream region of six genes that play a key role in DNA replication and repair: In these genes, PfH3p-HA demarcates a specific 1.5 kb chromatin island adjacent to the open reading frame. Our results indicate that, in *P. falciparum*, the process of histone clipping may precede chromatin integration hinting at preferential targeting of pre-assembled PfH3p-containing nucleosomes to specific genomic regions. The discovery of a protease-directed mode of chromatin organization in *P. falciparum* opens up new avenues to develop new anti-malarials.

**Keywords** epigenetics; histone H3 clipping; malaria
**Subject Categories** Chromatin, Epigenetics, Genomics & Functional Genomics; Microbiology, Virology & Host Pathogen Interaction

## Introduction

Malaria, a vector-borne disease transmitted by female *Anopheles* mosquitoes, still kills a child every minute [1]. The most severe form of human malaria is caused by the unicellular protozoan parasite *Plasmodium falciparum*, with clinical malaria symptoms manifesting during the 48-h asexual blood-stage cycle of the parasite. In the absence of efficient vaccines and the increasing spread of parasite drug resistance, treatment failures present a huge problem for malaria control and elimination. Over the past two decades, contributions to basic malaria research, particularly in the field of gene regulation, have identified new promising targets for anti-malarial drug development. For example, several studies have demonstrated a central role for post-translational modifications (PTMs) of histone N-terminal tails in regulating key *P. falciparum* intra-erythrocytic biology, antigenic variation, malaria pathogenesis, development of sexual stages that are transmitted to the vector, and virulence gene expression [2–7]. Accordingly, small molecules targeting these PTMs are in the pipeline for anti-malarial development [8–10].

As observed in virtually all eukaryotes, the nucleosome is the basic unit of chromatin structure in *P. falciparum* and comprises an octamer of core histones, around which are wrapped 147 bp of DNA. *Plasmodium falciparum* has four canonical core histones H2A, H2B, H3, and H4, and four histone variants H2A.Z, H2Bv, H3.3, and CenH3 [11]. *Plasmodium falciparum* also has a rich complement of chromatin-modifying proteins, and in fact, mass spectrometric analyses have shown that *Plasmodium* histones carry more than 60 PTMs, including acetylation, methylation, and phosphorylation [12–14]. However, only a few have been studied in depth. Much of our knowledge about the role of PTMs in

1  Departamento de Biomedicina Molecular, Centro de Investigación y de Estudios Avanzados del Instituto Politécnico Nacional (IPN), Ciudad de Mexico, México
2  Unité Biologie des Interactions Hôte-Parasite, Département de Parasites et Insectes Vecteurs, Institut Pasteur, Paris, France
3  CNRS, ERL 9195, Paris, France
4  INSERM, Unit U1201, Paris, France
5  Departamento de Biotecnología y Bioingeniería, Centro de Investigación y de Estudios Avanzados del Instituto Politécnico Nacional (IPN), Ciudad de México, México
   *Corresponding author. Tel: +33 145688616; E-mail: artur.scherf@pasteur.fr
   **Corresponding author. Tel: +52 55 5747 3325; E-mail: rohernan@cinvestav.mx
   †Present address: Institute of Bioinformatics and Applied Biotechnology, Bengaluru, India
   In memory of Dr. Guillermo Mendoza Hernández

*P. falciparum* gene regulation comes from the investigation of the clonally variant expression of the *var* gene family [7,15] and, more recently, the sexual commitment mechanism via variegated expression of a master regulator called AP2-G, a transcription factor of the ApiAP2 family [16,17].

In contrast to these reversible chemical modifications, it has recently emerged that proteolysis of histone tails is a type of irreversible PTM in eukaryotes, which can be resolved only by histone turnover or nucleosome remodeling. The consequences of histone tail processing include cell cycle progression, organismal development, viral infection, and aging [18,19]. For example, in mouse embryonic stem cells, it was shown that the clipping of the tail of histone H3 regulates cell differentiation [20]. A second study identified an endopeptidase that cleaves the tail of histone H3 in *Saccharomyces cerevisiae* and showed that the processed form of histone H3 controls the induction of gene expression by potentially clearing repressive PTMs [21]. Finally, a recent study demonstrated that mast cell lineage is governed by the tryptase-mediated clipping of histone H3 and H2B tails [22]. Thus, the biological outcome of cleavage of histone H3 differs in different organisms and cell types indicating that the clipped form is not just an intermediate of protein turnover.

In this work, we identified for the first time in a protozoan pathogen the clipping of the N-terminal region of histone H3 at amino acid 21, deleting the N-terminal tail from amino acids 1–21: This region is highly methylated and acetylated at positions lysine 4, lysine 9, lysine 14, and lysine 18, with particular marks being associated with transcriptional activation (H3K4me3 and H3K9ac) and others with regulation of variegated gene expression (H3K9me3), including of virulence genes involved in immune evasion and genes regulating sexual commitment [4,5,7,15,17, 23,24]. We also show that clipped histone H3 primarily integrates into chromatin regions upstream of six DNA replication gene loci hinting at the existence of a highly specific cellular targeting machinery for truncated histones. Overall, our data identify a novel epigenetic mechanism employed by *P. falciparum* that is linked to DNA metabolism.

## Results

### The N-terminal region of histone H3 is clipped at amino acid position 21 in *Plasmodium falciparum* intra-erythrocytic stages

To determine whether histone proteolysis occurs during *P. falciparum* intra-erythrocytic development, we prepared nuclear and cytoplasmic extracts of 3D7 parasites synchronized at the ring [6–10 hours post-invasion (hpi)], trophozoite (26–30 hpi), or schizont (36–40 hpi) stages, and analyzed them by immunoblotting with antibodies targeting the C-terminus of histone H3 or histone H4 (Fig 1A). The resulting pattern consisted of a 17 kDa band corresponding to full-length histone H3 (PfH3), a minor 14.5 kDa band, referred to here as the intermediate form PfH3int, and a 14 kDa terminally processed form PfH3p that was most prominent in schizonts (Fig 1A). In contrast, histone H4 migrated as a single band in all stages. Notably, we observed that the truncated forms of PfH3 are part of mononucleosomes prepared from the schizont stage (Fig 1B) and are not recognized by antibodies targeting the

N-terminus of histone H3 (Appendix Fig S1). These data indicate that N-terminal processing of *P. falciparum* histone H3 to H3p occurs during blood-stage development and that the processed form is incorporated into nucleosomes.

Next, to determine the position at which histone H3 truncation occurs, we recovered the 17 and 14 kDa bands from the polyacrylamide gel of Fig 1B and analyzed them by mass spectrometry. This analysis not only confirmed that the 14 kDa band was indeed a truncated form of histone H3 (Table EV1) and not the variant H3.3, but also identified previously known [12] and new histone PTMs in *P. falciparum* (mainly methylation of lysines and arginines; Table EV2). To further validate this finding, we probed schizont-stage mononucleosomes using antibodies that recognize different N-terminal histone H3 PTMs and found that PfH3p contained acetylated lysine 23 (H3K23ac) and tri-methylated lysine 27 (H3K27me3) marks but not H3K9me3 or H3K18ac (Fig 1C). Together, these results predicted a cleavage site between amino acids 18 and 23. Finally, to identify the exact position of PfH3 cleavage, we analyzed PfH3p using Edman degradation. This analysis showed that the cleavage site is located between amino acids 21 and 22 (Fig 1D).

### The protease inhibitor profile of an *in vitro* histone H3 proteolysis assay predicts a cathepsin C-like nuclear protease as being responsible for clipping

To identify the class of protease(s) involved in H3 processing, we developed an *in vitro* proteolysis assay using recombinant *P. falciparum* histone H3 fused to glutathione S-transferase at the N-terminus (GST-PfH3) as substrate, and schizont-stage nuclear extracts as the source of the proteolytic enzyme (Fig 2A); recombinant *P. falciparum* histone H2A tagged with GST (GST-PfH2A) served as a negative control. As shown in Fig 2B, incubation of GST-PfH3 (~ 43 kDa) with increasing amounts of nuclear extracts resulted in a truncated version GST-$NH_2$-H3 (~ 28 kDa), which migrated faster than the full-length protein on a polyacrylamide gel. In contrast, GST-PfH2A was not processed (Fig 2B, right). Subsequently, to determine the nature of the PfH3 protease, we performed our *in vitro* proteolysis assay in the presence of chemical inhibitors targeting different protease families including cysteine proteases, metalloproteases, aminopeptidases, or serine–cysteine proteases (Fig 2C and D). Only chymostatin, an inhibitor of serine–cysteine proteases of the cathepsin A, B, C, D, H, and L family, blocked GST-PfH3 processing (Fig 2C and D). However, because E64 (an inhibitor of cathepsin B, H, and L) and antipain (an inhibitor of cathepsin A, B, and D) did not abrogate GST-PfH3 processing, we infer by elimination that a cathepsin C-like protease found in the parasite nucleus may control the processing of histone H3 in intra-erythrocytic stages. A similar prediction was made using the peptide cutter server (http://www.dmbr.ugent.be/prx/bioit2-public/SitePrediction/), which predict that the cleavage site of *P. falciparum* histone H3 occurs at leucine 21 and that this cleavage could be performed by cathepsin C.

### Ectopically expressed clipped H3 integrates into nuclear mononucleosomes

We next investigated if histone H3 clipping occurs in *cis*-, i.e., in the context of the nucleosome, or in *trans*-, i.e., the clipped form of

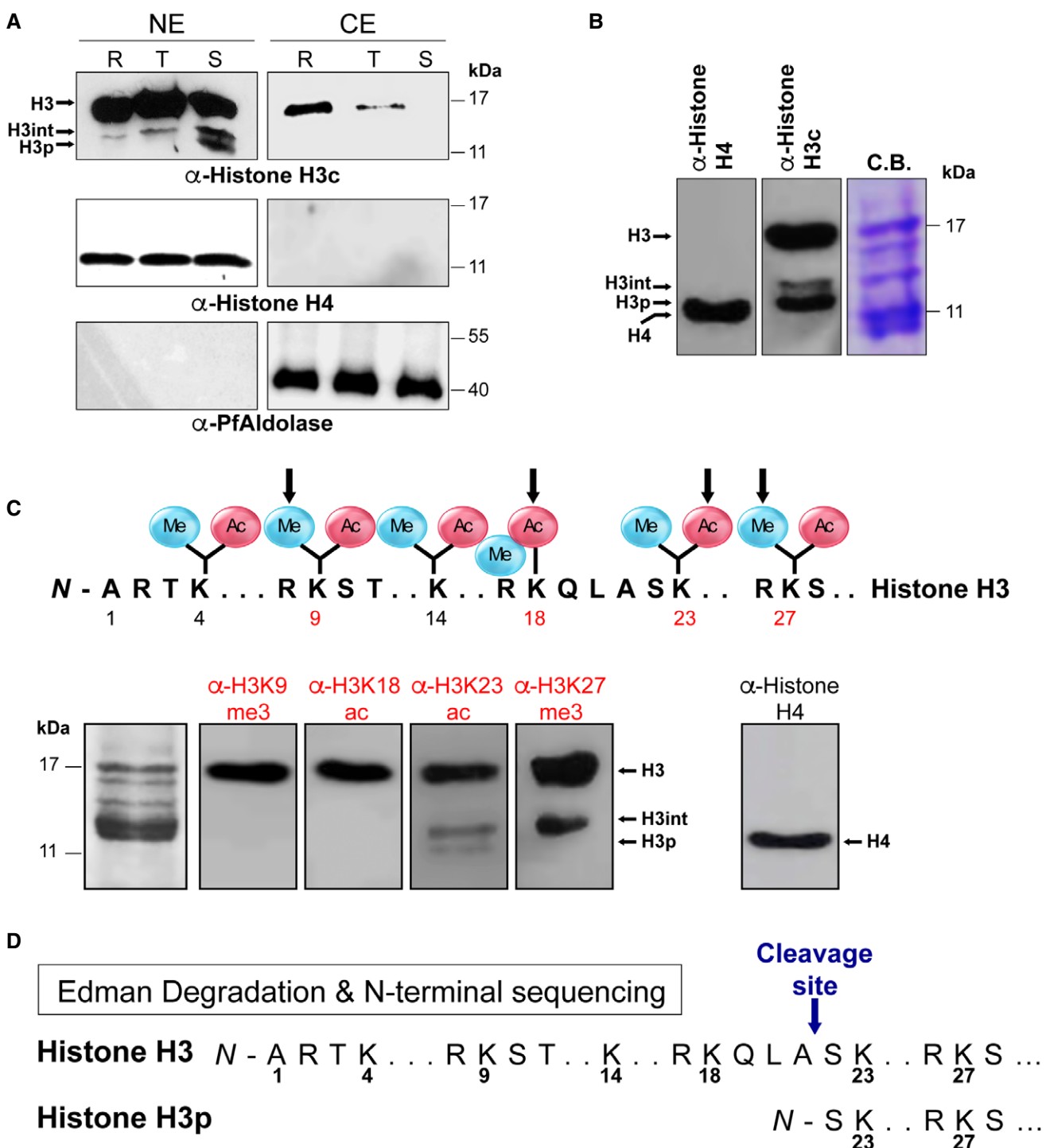

**Figure 1. *Plasmodium falciparum* nucleosomal histone H3 is proteolytically processed in a stage-specific manner between amino acids 21 and 22.**

A    Immunoblot analysis of nuclear and cytoplasmic extracts prepared from parasites synchronized at ring (R), trophozoite (T), and schizont (S) stages with anti-histone H3 C-terminus antibodies identifies three bands: full-length histone H3, an intermediate form (H3-int), and a fully processed form (H3p). Anti-histone H4 and anti-PfAldolase antibodies served as loading controls for the nuclear and cytoplasmic fraction, respectively.

B    Immunoblot assays of mononucleosomes with anti-histone H3 C-terminus or anti-histone H4 antibodies. The right panel shows a mononucleosome preparation separated on a denaturing polyacrylamide gel and stained with Coomassie Brilliant Blue (C.B.).

C    Immunoblot analysis of a schizont-stage histone preparation with antibodies against the indicated post-translational modification (PTM) in the N-terminal tail of histone H3 (upper panel).

D    Results from the Edman degradation analysis of full-length PfH3 or processed PfH3p; the arrow indicates the position of histone H3 clipping.

Source data are available online for this figure.

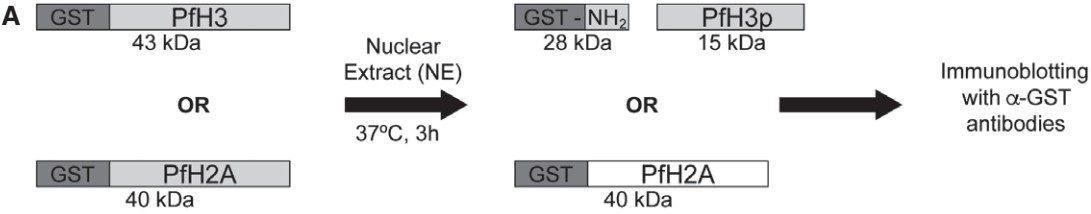

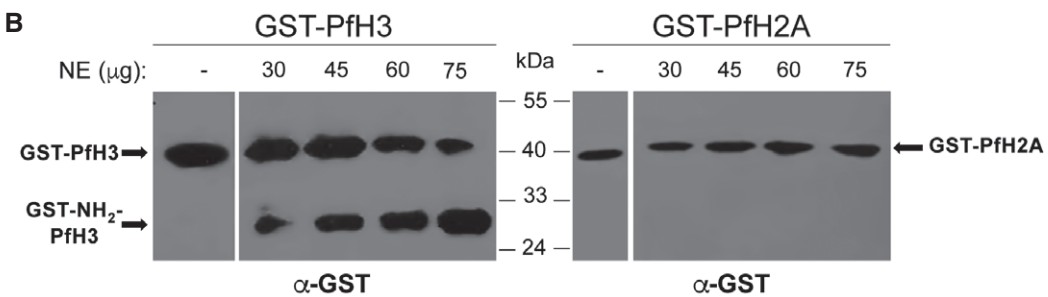

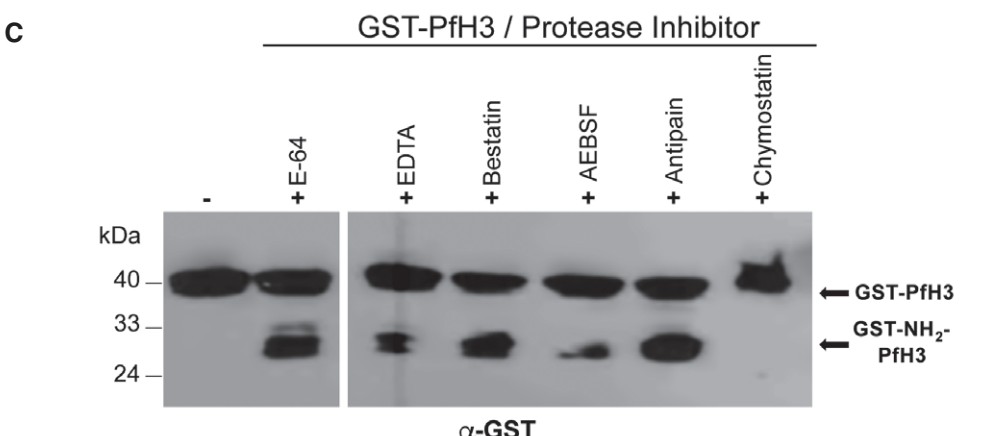

| Inhibitor | Specificity | Inhibition |
|---|---|---|
| E-64 | Cysteine proteases (Cathepsin B, H, L) | - |
| EDTA | Metalloproteases | - |
| Bestatin | Aminopeptidases | - |
| AEBSF | Serine proteases | - |
| Antipain | Serine cysteine proteases (Cathepsin A,B,D) | - |
| Chymostatin | Serine cysteine proteases (Cathepsin A,B,C,D,H,L) | ++ |

**Figure 2. A cathepsin C-type protease in schizont-stage nuclear extracts may mediate *Plasmodium falciparum* histone H3 processing.**

A Schematic representation of the *in vitro* proteolysis assay developed in this study. N-terminally GST-tagged PfH3 was incubated with schizont-stage nuclear extracts at 37°C for 3 h and analyzed by immunoblotting with anti-GST antibodies. GST-tagged PfH2A served as a control.

B GST-PfH3 or GST-PfH2A was incubated with increasing amounts of nuclear extracts, and the reactions were resolved using denaturing gel electrophoresis and analyzed by immunoblotting with anti-GST antibodies.

C GST-PfH3 was incubated with 25 μg of nuclear extracts and the indicated protease inhibitor. The resulting products were analyzed by denaturing gel electrophoresis and immunoblotting with anti-GST antibodies. (−) indicates a reaction without nuclear extract and protease inhibitor.

D Target profile of the protease inhibitors used in panel (C).

Source data are available online for this figure.

histone H3 could be integrated into nucleosomes by chromatin remodeling enzymes. To distinguish between these models, we expressed PfH3p ectopically from an episome with a C-terminal triple-HA tag under the control of the weak CRT promoter (see Material and Methods for more information) and confirmed PfH3p-HA expression in our transgenic parasites (Appendix Fig S2). Thereafter, we performed immunofluorescence assays on fixed asexual blood-stage parasites using anti-HA antibodies and found that the PfH3p-HA signal was restricted to the nucleus in all three developmental stages, the rings, trophozoites, and schizonts (Fig 3A). Furthermore, using purified mononucleosomes from wild-type (WT) and transgenic (WT + PfH3p-HA) parasites synchronized at the schizont stage, we found that PfH3p-HA forms an integral part of the nucleosomes (Fig 3B). Importantly, the clipping of endogenous histone H3 is not lost in the transgenic parasites (Fig 3B, lanes labeled H3c). Lastly, immunoprecipitation experiments of purified mononucleosomes obtained from WT or WT + PfH3p-HA schizont-stage parasites with anti-HA antibodies co-immunoprecipitated histone H4 whereas the mouse IgG control did not (Fig 3C). These data indicate that ectopically expressed PfH3p-HA interacts normally with canonical histones in the context of nucleosomes supporting a *trans-* model for clipped histone H3 incorporation into *P. falciparum* chromatin.

**Clipped histone H3 is targeted to the 5′UTR of genes regulating DNA replication**

Based on the above observation, we decided to use our transgenic parasite line to identify genomic regions that are targeted by clipped histone H3. Using anti-HA antibodies, which immunoprecipitate PfH3p-HA, and anti-N-terminal histone H3 antibodies, which immunoprecipitate histone H3 with an intact N-terminal tail (referred to hereafter as PfH3n), we performed chromatin immunoprecipitation (ChIP) followed by next-generation sequencing (NGS) (-seq) of schizont-stage WT + PfH3p-HA parasites. Downstream differential peak-calling analysis was performed using MACS2 [25]: Comparisons included PfH3p-HA signal relative to the ChIP input, PfH3p-HA signal relative to the PfH3n, and PfH3n enrichment over ChIP input.

First, we visualized the genome-wide distribution of the anti-HA ChIP-seq signal (labeled PfH3p-HA) and its fold enrichment (FE) relative to either the ChIP input (labeled PfH3p-HA/input) or PfH3n (labeled PfH3p-HA/PfH3n) using custom scripts and the IGV browser [26], and observed that fewer than 20 genomic regions were enriched for PfH3p-HA across all 14 *P. falciparum* chromosomes (Fig 4A). The clustering of biological replicates for each ChIP-seq condition in a principal component analysis (Fig 4B), and the Pearson correlation coefficients between replicates in ChIP-seq as well as for MACS2-derived FE profiles (Appendix Fig S3), shows a high degree of consistency between replicates. Next, using MACS2-based peak-calling analysis, we identified 11 peaks that overlapped between the PfH3p-HA versus input comparison (Table EV3A) and the PfH3p-HA versus PfH3n comparison (Dataset EV1A), at a cutoff of 10 FE and a *P*-value of 0.01 after adjusting for the false discovery rate [27]. These 11 peaks correspond to 12 genes that are listed in Table 1 along with the position of the peak relative to the coding sequence (cds). Of note, six of the 12 H3p-enriched genes encode for proteins that

regulate DNA replication and repair: replication protein A type 1 (RPA1), proliferating cell nuclear antigen 1 (PCNA1), single-stranded DNA-binding protein (SSB), topoisomerase I (TOPO-I), topoisomerase II (TOPO-II), and DNA polymerase alpha subunit (DNA pol α). Importantly, the H3p signal for these regions always begins within the 5′UTR and spans a region of approximately 1.5 kb, including up to 100 bp of the cds (Fig 4D and Appendix Fig S4, and Table 1). We also performed gene ontology (GO) analysis of the PfH3p-HA enriched genes (Table EV3B and Dataset EV1A) and identified GO terms related to DNA replication, DNA binding, and DNA-dependent functionality (e.g., regulation of DNA metabolic process, DNA topological change, DNA replication, DNA topoisomerase activity, DNA replisome, and replication fork) as being enriched.

To determine whether the observed PfH3p-HA localization pattern was parasite stage-specific, we performed ChIP-seq analysis of ring-stage WT + PfH3p-HA parasites using anti-HA and anti-PfH3n antibodies and observed a strong correlation between the schizont and ring biological replicates for PfH3p-HA (Appendix Fig S5). Furthermore, MACS2 peak-calling analysis identified only six peaks that overlapped between the PfH3p-HA versus input comparison (Dataset EV2) and the PfH3p-HA versus PfH3n comparison (Dataset EV3), at a cutoff of 10 FE and *P*-value of 0.01 after adjusting for false discovery rate. These corresponded to the six replication genes RPA1, PCNA1, SSB, TOPO-I, TOPO-II, and DNA pol α (Appendix Fig S6). Taken together, our ChIP-seq data further support a *trans-* model of histone H3p incorporation into nucleosomes and reveal the existence of a novel mechanism in *P. falciparum* that can recruit pre-existing clipped histone H3 to specific genomic loci.

# Discussion

Here, we report a novel type of histone PTM in malaria parasites that involves a nuclear protease. We observe histone H3 clipping during asexual blood development of *P. falciparum* generating two closely related forms of truncated H3, PfH3int and PfH3p. Although we call PfH3int the intermediate form, it may represent the same truncated form as PfH3p but with distinct PTM status, which is supported by the detection of the doublet by anti-H3K23ac antibodies but not by anti-H3K27me3 antibodies (see Fig 1C). We also demonstrate that only a small number of genes have nucleosomes that contain clipped PfH3p-HA. Strikingly, the majority of genes that show enriched PfH3p-HA in their 5′UTR region are associated with DNA metabolism. We speculate that *P. falciparum* DNA replication genes are marked by this truncated form of histone H3 to provide a specific mechanism to coordinate DNA synthesis and subsequent karyokinesis during the process of parasite multiplication, i.e., schizogony. The observed association may therefore represent a malaria parasite-specific adaptation that does not exist in other eukaryotic species. It will be interesting to see whether other closely related Apicomplexan parasites such as *Toxoplasma* spp. and *Babesia* spp. utilize a similar regulatory mechanism.

Histone H3 N-terminal truncation at different sites has been reported to permanently remove epigenetic histone PTMs in several eukaryotes (for a review, see ref. [19], which in turn results in histone turnover or gene regulation). In these published cases,

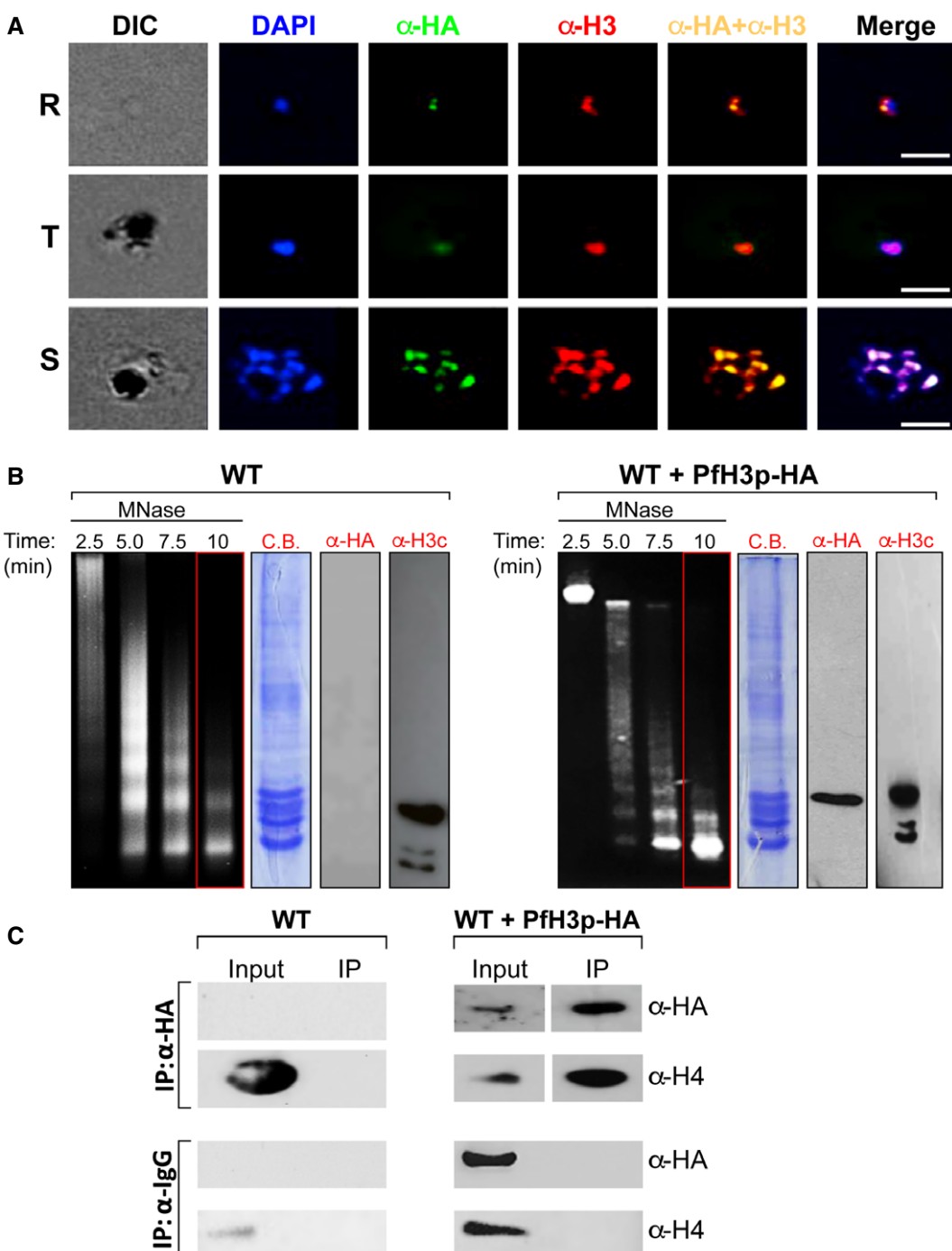

**Figure 3. Ectopically expressed histone H3p localizes to the nucleus during parasite asexual development and incorporates into nucleosomes.**

A Indirect immunofluorescence assays were performed to determine the localization of ectopically expressed PfH3p-HA in ring (R), trophozoite (T), and schizont (S) stages of *Plasmodium falciparum* asexual growth. PfH3p-HA was detected using anti-HA antibodies (green) and endogenous histone H3 with anti-histone H3 N-terminal antibodies (red). DAPI (blue) was used to stain the nucleus. Scale bar = 5 μm.

B Nuclei isolated from wild-type (WT) or PfH3p-HA-expressing (WT + PfH3p-HA) schizont-stage parasites were treated with 4 U/ml of micrococcal nuclease (MNase) for the indicated amounts of time, the DNA purified and migrated on a 2% agarose gel, and stained with ethidium bromide. Mononucleosomes purified after 10 min of MNase treatment were separated using denaturing polyacrylamide gel electrophoresis and either stained with Coomassie Brilliant Blue (C.B.) or visualized by immunoblotting with anti-HA (α-HA) or anti-C-terminal histone H3 (α-H3c) antibodies.

C Co-immunoprecipitation (IP) experiments of purified mononucleosomes obtained from wild-type (WT) or transfected (WT + PfH3p-HA) schizont-stage parasites were performed with either anti-HA antibodies or mouse IgG. Immunoprecipitated products (right panel) were analyzed by immunoblotting using anti-HA or anti-histone H4 antibodies.

Source data are available online for this figure.

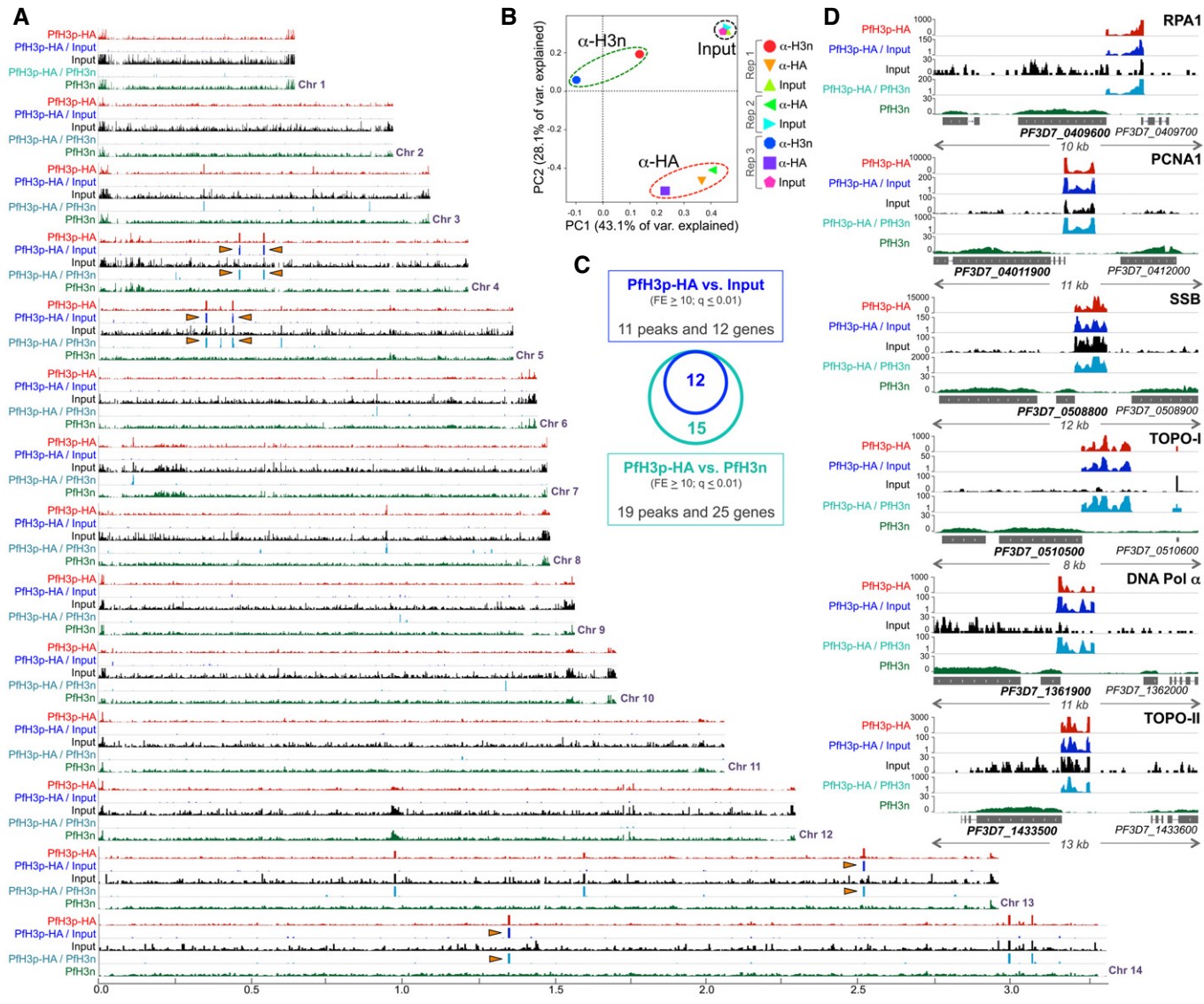

**Figure 4.  In *Plasmodium falciparum*, clipped histone H3 is targeted to the 5′UTR of genes regulating DNA replication.**

A   Genome-wide distribution of ectopically expressed PfH3p-HA in *P. falciparum* schizont stages is represented as fold enrichment of PfH3p-HA ChIP-seq signal over input (PfH3p-HA/input in blue; *y*-axis scale 1–10) or over PfH3n (PfH3p-HA/input in teal; *y*-axis scale 1–10) calculated using MACS2. The coverage of PfH3p-HA (red; *y*-axis scale 0–50), input (black; *y*-axis scale 0–30), and PfH3n (green; *y*-axis scale 0–30) is also shown and represents average reads per million over 1,000 nt bins of the genome. The *x*-axis represents chromosome size in Mb. Major MACS2-derived peaks identified in all replicates are indicated using an orange arrowhead.

B   Principal component analysis of the different ChIP-seq replicates (bigwig files derived from deduplicated bam files) was performed using the plotPCA function of deepTools on a multibigwig summary file, over 150 nt bins. The Eigenvalues of the top two principal components PC1 and PC2 are shown, and meaningful clustering of replicates is highlighted. Rep = biological replicate.

C   Summary of the PfH3p-enriched peaks—and corresponding genes—identified in the PfH3p-HA versus input or PfH3p-HA versus PfH3n comparisons using MACS2 peak-calling analysis. The overlap between the two comparisons is indicated using a Venn diagram. FE = fold enrichment; *q*-value represents the Benjamini and Hochberg false discovery rate.

D   The genomic context of the six peaks identified in all three PfH3p-HA ChIP-seq replicates is shown. The fold enrichment of PfH3p-HA ChIP-seq signal over input (PfH3p-HA/input; blue) or PfH3n (PfH3p-HA/PfH3n; teal) is shown, as are the coverage plots of PfH3p-HA (red), PfH3n (green), and input (black), which are represented as average reads per million over 1,000 nt bins of the genome. DNA replication genes: replication protein A 1 (RPA1), proliferating cell nuclear antigen 1 (PCNA1), single-stranded DNA-binding protein (SSB), topoisomerase I (TOPO-I), DNA polymerase alpha subunit (DNA pol a), and topoisomerase II (TOPO-II) are highlighted and their directionality indicated. Note that the data in part D correspond to Biological Replicate 1.

histone H3 clipping phenotypes were not linked to DNA replication blockage but affected, for example, differentiation of mouse embryonic stem cells [20] and the expression of stationary-phase genes in *S. cerevisiae* [21]. Furthermore, most of these studies did not even

consider if the protease could operate in *trans-*, the default assumption was that it acted in *cis-*. In contrast, our experimental data from malaria parasites support a cellular mechanism that incorporates ectopically clipped histone H3 into chromatin regions at particular

Table 1.   List of PfH3p-HA peaks and their *Plasmodium falciparum* genomic location.

| Peak_name[a] | Location | | | Length (bp) | Gene_ID (Strand[b]) | Position (5′UTR, cds, 3′UTR) | Gene product |
|---|---|---|---|---|---|---|---|
| | Chr. | Start | End | | | | |
| S.all.HAvI.2_peak_5 | 4 | 534,319 | 535,814 | 1,496 | PF3D7_0411900 (−) | cds, 5′UTR | **DNA polymerase alphasubunit** |
| | | | | | PF3D7_0412000 (+) | 5′UTR | LITAF-like zinc finger protein, putative |
| S.all.HAvI.2_peak_7 | 5 | 364,306 | 365,998 | 1,693 | PF3D7_0508800 (−) | cds, 5′UTR | **Single-stranded DNA-binding protein** |
| S.all.HAvI.2_peak_8 | 5 | 448,393 | 450,065 | 1,673 | PF3D7_0510500 (−) | cds, 5′UTR | **Topoisomerase I** |
| S.all.HAvI.2_peak_15 | 14 | 133,7159 | 133,8757 | 1,599 | PF3D7_1433500 (−) | cds, 5′UTR | **DNA topoisomerase 2** |
| S.all.HAvI.2_peak_14 | 13 | 248,3093 | 248,4802 | 1,710 | PF3D7_1361900 (−) | cds, 5′UTR | **Proliferating cell nuclear antigen 1** |
| S.all.HAvI.2_peak_4[c] | 4 | 456,175 | 457,042 | 868 | PF3D7_0409700 (−) | cds, 5′UTR | **Replication protein A1, large subunit** |
| | | | | | PF3D7_0409600 (+) | 5′UTR | Peptide chain release factor 2 |
| S.all.HAvI.2_peak_9 | 7 | 514,911 | 515,153 | 243 | PF3D7_0711700 (−) | cds | Erythrocyte membrane protein 1, PfEMP1 |
| S.all.HAvI.2_peak_11 | 9 | 103,795 | 104,047 | 253 | PF3D7_0902300 (−) | cds | Serine/threonine protein kinase, FIKK |
| S.all.HAvI.2_peak_13 | 13 | 973,880 | 974,032 | 153 | PF3D7_1323400 (−) | 5′UTR | 60S ribosomal protein L23 |
| S.all.HAvI.2_peak_1 | 2 | 275,544 | 275,782 | 239 | PF3D7_0206900 (−) | cds | Merozoite surface protein 5 |
| S.all.HAvI.2_peak_3[c] | 4 | 455,551 | 456,010 | 460 | PF3D7_0409600 (−) | cds, 5′UTR | **Replication protein A1, large subunit** |
| | | | | | PF3D7_0409700 (+) | 5′UTR | Peptide chain release factor 2 |

Bold are indicated the genes that encoded for proteins implicated in DNA replication and repair.
[a]Peak_name was derived from Table EV3A.
[b]Indicates the strand corresponding to the coding sequence.
[c]S.all.HAvI.2_peak_4 and S.all.HAvI.2_peak_3 localize to the same genomic region.

genome loci: *P. falciparum* chaperones specific to clipped histone H3 may target PfH3p-containing pre-assembled nucleosomes to specific gene loci or chromatin remodeling enzymes such as the SWI/SNF complex, and others [28] may exchange full-length histone H3 for PfH3p at replication gene loci. Indeed, in the absence of a selection process for clipped histones, we would have expected the ectopically expressed PfH3p-HA to be distributed throughout the *P. falciparum* genome. Nevertheless, we cannot rule out that histone H3 clipping occurs in the chromatin context: This would require the targeting of the histone H3 protease to specific genomic regions, which can be addressed upon protease identification.

Thus far, diverse families of proteases have been implied in histone H3 clipping, including cathepsin L-type cysteine proteases [20], tryptase [22], glutamate dehydrogenase [29], Jumonji C domain (JmjC)-containing proteins, JMJD5 and JMJD7 [30], and an unidentified yeast enzyme that has intrinsic aminopeptidase and endopeptidase activity [21]. Interestingly, Duncan *et al* [20] reported that covalent histone modifications modulate cathepsin L activity. Our data also point to a histone modification upstream of the truncated region at lysine 27 that may contribute to H3 N-terminal clipping. The cleaved H3 shows H3K27me3, a methylation mark that was previously not reported for malaria parasites. In *P. falciparum,* the utilization of a variety of protease inhibitors in the *in vitro* proteolysis assay hinted at the presence of a cathepsin C-type exo-cysteine peptidase in parasite nuclear extracts that cleaves histone H3 at its N-terminus. Interestingly, in T lymphocytes and natural killer cells, cathepsin C is a lysosomal protease that has been implicated in processing pro-granzymes into proteolytically active forms [31]. It also plays a role in epithelial differentiation [32], with genetic mutations of the cathepsin C gene CTSC causing the autosomal recessive disorder Papillon–Lefèvre syndrome [33].

The *P. falciparum* genome (www.plasmodb.org/plasmo) encodes three cysteine proteases of the cathepsin C subtype, also known as dipeptidyl aminopeptidase (DPAP). Two of them, PfDPAP1 (*PF3D7_1116700*) and PfDPAP3 (*PF3D7_0404700),* are expressed in the cytoplasm of asexual blood stages and are involved in vacuolar protein degradation and erythrocyte invasion, respectively [34,35], while PfDPAP2 (*PF3D7_1247800*) expression was observed in sexual-stage parasites [36]. Furthermore, a cathepsin L-like protease has not been annotated in the PlasmoDB genome database (https://plasmodb.org). Biochemical fractionation of nuclear extracts followed by mass spectrometry may reveal the nature of the candidate clipping protease, which is beyond the scope of this work.

In conclusion, the discovery of a novel type of PTM in malaria parasites predicts the existence of an added layer of epigenetic regulation in this highly complex pathogen. Given that the N-terminal region of histone H3 (amino acids 1–21) is a key regulator of gene expression in *P. falciparum*, especially of virulence genes, regulating the N-terminal tail by proteolysis may trigger a so-far unknown response, possibly cell cycle progression or overcoming a mitotic checkpoint to enable

karyokinesis. Furthermore, it opens up novel avenues for reducing malaria mortality. If histone clipping is conserved during *P. falciparum* proliferative stages in the mosquito and human liver, specific inhibitors of the clipping process may target pre- and intra-erythrocytic development as well as mosquito stages. Finally, the identification of the gene that codes for the nuclear clipping protease is of particular interest to investigate the biological role of histone clipping during the various developmental stages of this major human pathogen.

# Materials and Methods

### Parasite culture

*Plasmodium falciparum* parasites of the laboratory-adapted 3D7 strain were cultured following standard protocols [37]. Parasites were synchronized with 5% sorbitol and plasmion on a regular basis [38]. Transfection of ring-stage parasites of the 3D7 genotype using 100 μg of the pARL-H3K22-HA(3X) vector was performed as described previously [39]. Parasites carrying the pARL-H3K22-HA (3X) plasmid (described below) were selected with 10 nM WR99210 (Walter Reed Army Institute of Research).

### Generation of transgenic parasites

Transgenic parasites expressing a truncated version of *P. falciparum* histone H3 (amino acids S22–S136) tagged with three haemagglutinin A [HA(3X)] epitopes at its C-terminus were generated as follows. First, the vector pARL-SSG (in which the ORF of a Stevor gene is fused to green fluorescent protein (GFP) and expressed from the CRT promoter [40]) was digested with the *XhoI* and *KpnI* restriction enzymes to release the *Stevor* and GFP cds from the pARL vector backbone. Next, the H3S22-S136-HA(3X) cassette was amplified by PCR using genomic DNA from the wild-type 3D7 strain as template and the following sense and antisense primers:

*XhoI*-H3K21 (5′-CCGCTCGAGTCAAAGCAGCAAGGAAATCA-3′) and *KpnI*-HA(3X) (5′-GGGGTACCTTATGCATAATCTGGTACATCAT ATGGATATGCATAATCTGGT ACATCATATGGATATGCATAATCTG GTACATCATATGGATAAGATCTTTCCCACGAATACG-3′) to amplify the truncated version of histone H3. This insert was digested with *XhoI* and *KpnI* enzymes and ligated to the pARL vector backbone to obtain the transfection vector pARL-H3S22-HA(3X), which was amplified for transfection into 3D7. Drug-resistant parasites emerged after 4 weeks of continuous culture in the presence of 5 nM WR92210.

### Nuclear and cytoplasmic extract preparation

Nuclear and cytoplasmic extracts from synchronized asexual stages (ring, trophozoites, and schizonts) were prepared as described previously [41]. In order to prepare nuclear extracts for use in the *in vitro* proteolysis assays, buffers were *not* supplemented with protease inhibitors.

### Mononucleosome preparation

Mononucleosome preparation was carried out as described previously [41]. Briefly, schizont-stage parasites of WT or WT + PfH3p-HA were isolated from infected erythrocytes by saponin lysis, resuspended in 1 ml of buffer A (10 mM HEPES, pH 7.9, 10 mM KCl, 1.5 mM MgCl$_2$) supplemented with protease inhibitors (Complete, Roche), and incubated for 30 min at 4°C. Parasites were lysed with 200 strokes in a prechilled Dounce homogenizer and nuclei collected by centrifugation, cleaned through a 0.34 M sucrose cushion, and resuspended in 1 ml of prechilled buffer A supplemented with 2 mM CaCl$_2$ plus protease inhibitors. Mononucleosomes were prepared from freshly isolated nuclei by digestion with 4 U MNase (NEB) for different amounts of time at 33°C; digestion was stopped by adding 10 mM EGTA pH 8.0. After spinning out debris at 18,800 *g* for 15 min at 4°C, the supernatant was analyzed for the presence of mononucleosomes by extracting DNA and resolving it on a 2% agarose gel stained with ethidium bromide. Typically, after 30 min of MNase digestion, the population contained 95% mononucleosomes.

### Antibodies and immunoblotting

The following commercial antibodies (Ab) were used for immunoblotting assays: Anti-histone H3 C-terminal Ab (Abcam Ab1791) were used at a 1:150,000 dilution, anti-histone H3 N-terminal Ab (Sigma-9289) were used at 1:1,000 dilution, anti-histone H4 Ab (Santa Cruz sc-8658-R) were used at 1:3,000 dilution, anti-H3K9me3 Ab (Millipore 07-442) were used at a 1:3,000 dilution, anti-H3K18ac Ab (Millipore 07-354) were used at a 1:40,000 dilution, anti-H3K23ac Ab (Abcam ab46982) were used at a 1:100,000 dilution, anti-H3K27me3 Ab (Abcam ab6147) were used at a 1:2,000, anti-H3K27ac Ab (Millipore 07-360) were used at a 1:40,000 dilution, anti-GST Ab (Invitrogen 13-6700) were used at a 1:50,000 dilution, and anti-HA Ab (Roche 12CA5) were used at a 1:2,000 dilution. After incubation with horseradish peroxidase-conjugated secondary antibodies, membranes were developed with SuperSignal West Pico Chemiluminescent Substrate (Pierce) according to the manufacturer's instructions.

### Histone extraction and mapping of H3 cleavage site

Acid extraction of histones from synchronized asexual stages (ring, trophozoites, and schizonts) of *P. falciparum* was carried out as previously described [41]. In brief, infected erythrocytes were harvested by centrifugation at 500 *g* for 5 min and parasites isolated by saponin lysis. The resulting parasite pellet was resuspended in 1 ml of lysis buffer (10 mM Tris pH 8, 15 mM KCl, 1.5 mM MgCl$_2$, 1 mM DTT, 0.25% NP-40) supplemented with protease inhibitors (Complete, Roche) and incubated for 30 min at 4°C. Parasites were then lysed with 200 strokes in a prechilled Dounce homogenizer, and the nuclei were collected by centrifugation at 9,600 *g* for 15 min and washed once with 10 volumes of 0.3 M NaCl and twice with 10 volumes of 0.5 M NaCl. Histones were extracted by incubating the samples with 10 volumes of 0.25 M HCl overnight at 4°C. The extracts were centrifuged at 18,800 *g* for 30 min to remove insoluble debris and histones precipitated with eight volumes of cold acetone (incubation overnight at −20°C) and centrifugation at 13,800 *g* for 30 min at 4°C. The final pellets were air-dried, resuspended in 50 mM Tris–HCl pH 8.8, and stored at −80°C.

To determine the cleavage site of PfH3, equal amounts of acid-purified *P. falciparum* histones were separated by 18% SDS–PAGE and transferred onto nitrocellulose membranes. Membranes were probed with commercially available antibodies against core histone H4, core histone H3 (C-terminal), H3K9me3, H3K18ac, H3K23ac,

H3K27me3, and H3K27ac. Alternatively, histone H3 and its processed forms were excised from the nitrocellulose membrane, after probing with core histone H3 antibodies, and subjected to Edman degradation as described previously [42].

## Mass spectrometry

In-gel digestion of proteins was carried out according to the manufacturer's manual (Pierce, Rockford, IL). Briefly, the proteins of interest were separated on a denaturing polyacrylamide gel and stained with Coomassie blue. The band of interest was excised, destained, and disulfide bonds reduced with Tris 2-carboxyethylphosphine (TCEP) and alkylated with iodoacetamide. The gel pieces were then dehydrated with acetonitrile and rehydrated with 100 ng trypsin (Promega, Madison, WI) in 25 mM ammonium bicarbonate solution and incubated at 37°C overnight. The tryptic fragments were extracted from the gel by adding 1% trifluoroacetic acid. Next, the peptides were purified with a C18 reversed-phase minicolumn filled into a micropipette tip, i.e., ZipTip C18 (Millipore, Bedford, MA). Purified peptides were co-crystallized with α-cyano-4-hydroxycinnamic acid matrix (Applied Biosystems, Foster City, CA, USA) on a matrix-assisted laser desorption ionization (MALDI) target plate. Both mass spectrometry (MS) and MS/MS spectra were acquired in a MALDI–time-of-flight (MALDI–TOF/TOF) mass spectrometer (Applied Biosystems 4800 Proteomics Analyzer) in the reflector mode. Calibration was updated before each acquisition using a standard peptide mixture according to the instrument's protocol. Protein identification was performed using the GPS Explorer software (Applied Biosystems) with the MASCOT search engine (Matrix Science). The search was performed against the NCBI-nr 20080502 database (6493741 sequences; 2216039219 residues), and the Mascot significance threshold was set at $P = 0.05$. Individual ions scores > 33 indicated identity or extensive homology and were considered to be significant.

## Tandem Mass Spectrometry Analysis (LC/ESI-MS/MS) to identify post-translational modifications of histones

Protein bands were carefully excised from a Coomassie-stained SDS–polyacrylamide gel (SDS–PAG) and destained with 50% (v/v) methanol, 5% (v/v) acetic acid for 12 h. The destained gels were washed with deionized water, soaked for 10 min in 100 mM ammonium bicarbonate, cut into small pieces, dehydrated with 100% acetonitrile, and vacuum-dried. Proteins were reduced with 10 mM DTT and S-alkylated cysteine with 100 mM iodoacetamide in 100 mM ammonium bicarbonate. In-gel digestion was performed by adding 30 μl of modified porcine trypsin solution (20 ng/μl) in 50 mM ammonium bicarbonate followed by overnight incubation at room temperature. Peptides were extracted with 50% (v/v) acetonitrile, 5% (v/v) formic acid twice for 30 min each with sonication. The extracts were dried under vacuum and resuspended in 20 μl of 0.1% formic acid.

HPLC/MS/MS analysis of peptides was carried out using an integrated nano-LC-ESI_MS/MS system: It consists of a nanoACQUITY ultraperformance liquid chromatography (UPLC; Waters Corporation) coupled to a Q-ToF Synapt G2 High Definition Mass Spectrometer (Waters Corporation) equipped with a NanoLockSpray ion source. The mass spectrometer was calibrated with a NaCsI solution

(mass range: 50–2,000 Da) and operated in ESI positive V-mode at a resolution of 10,000 fwhh (full with at half height). Spectra were acquired in automated mode using data-dependent acquisition (DDA). [Glu[1]]-fibrinopeptide B solution (100 fmol/μl) was infused through the reference sprayer of the NanoLockSpray source at a flow rate of 0.5 μl/min and was sampled at 30-s intervals during the acquisition. MS survey scans of 1 s over the $m/z$ range 300–1,600 were used for peptide detection followed by two MS/MS scans of 2 s each ($m/z$ 50–2,000) of detected precursors. Collision energies were automatically adjusted based upon the ion charge state and the mass. The five most intense precursor ions of charge $2^+$, $3^+$, or $4^+$ were interrogated per MS/MS switching event. Dynamic exclusion for 60 s was used to minimize multiple MS/MS events for the same precursor. The data were post-acquisition lock mass corrected using the doubly protonated monoisotopic ion of [Glu[1]]-fibrinopeptide.

Data-dependent acquisition raw data files were processed and searched using the ProteinLynx Global Server, version 2.4, software (PLGS; Waters Corporation). The default parameter settings included: perform lock spray calibration with a lock mass tolerance of 0.1 Da, background substract type of adaptive, and deisotoping type of medium. PLGS was configured to automatically output PKL files which were subsequently database searched by MASCOT search algorithm (version 1.6b9, Matrix Science, London, UK) available at http://www.matrixscience.com. Mascot searches were conducted using the *P. falciparum* subset (Feb 2011, 16,710 sequences) of the National Center for Biotechnology Information non-redundant database (NCBI-nr, http://www.ncbi.nih.gov). Trypsin was set as the digest reagent, and one missed cleavage site was allowed. Mass tolerances of 30 ppm and 0.6 Da were used for precursor and product ions, respectively. Fixed modifications: carbamidomethyl-cysteine. Variable modifications: oxidation (M), deamidation (Q, N), acetylation (K, N-terminal), methylation, and dimethylation (K). Peptide matches with Mascot scores exceeding the 95% level of confidence were accepted as correct matches. Ion score is $-10*\text{Log}(P)$ where $P$ is the probability that the observed match is a random event. The threshold score was 40 for $P < 0.05$.

## Purification of recombinant PfH3 and PfH2A histones

The PfH3 ORF (*PF3D7_0610400*; START to STOP codon = 408 bp) was PCR-amplified using the following primers: PfH3-*EcoRI* (5′-GGAATTCCATGGCAAGAACTAAACAAACA-3′) and PfH3-*XhoI* (5′-CCGCTCGAGTGATCTTTCTCCACGGATACG-3′) and inserted into the pGEX-T1 vector (GE Healthcare), downstream of the glutathione S-transferase (GST) sequence. Similarly, the ORF of PfH2A (*PF3D7_0617800*; START to STOP codon = 396 bp) was PCR-amplified using the following primers: PfH2A-*EcoRI* (5′-GGAATTCCATGTCAGCAAAAGGAAAAACT-3′) and PfH2A-*XhoI* (5′-CCGCTCGAGTTAATAATCTTGATTGGCAGT-3′) and inserted into pGEX4-T1, downstream of the GST-coding sequence. The resulting constructs were transformed into *Escherichia coli* DH5α cells and grown at 30°C in Luria Bertani (LB) medium with 100 μg/ml ampicillin, under good aeration to an OD of 0.4–0.6, at which point GST-tagged protein expression was induced with 2 mM IPTG for 2 h at 30°C. Induced cells were harvested by centrifugation at 2,400 *g* for 10 min. The cell pellets were resuspended in PBS, disrupted by sonication on ice and centrifuged for 20 min at 6,900 *g* to get rid of unbroken cells and

other debris. GST-tagged proteins were purified using Glutathione Sepharose 4B beads (GE Healthcare) according to the manufacturer's instructions and eluted in 300 μl buffer. Protein concentration was determined using a Bradford assay (24). 1–2 μg of GST-PfH3 or GST-PfH2A was resolved using 12% SDS–PAGE, and protein purity was determined by Coomassie blue staining.

### *In vitro* proteolysis assays

*In vitro* proteolysis assays were performed as described previously [20]. Briefly, 45 μg of *P. falciparum* nuclear extract (as a source of proteases) was mixed with 1.5 μg of recombinant protein (GST-PfH3 or GST-PfH2A) in a 60 μl reaction volume in a buffer containing 10 mM HEPES, 10 mM KCl, 1.5 mM $MgCl_2$, 0.34 M sucrose, 10% glycerol, and 5 mM β-mercaptoethanol and incubated for 3 h at 37°C. The resulting products were resolved on a 12% SDS–PAGE gel and the proteolytic pattern analyzed by immunoblotting with HRP-conjugated anti-GST antibodies. For Fig 2D, inhibitors specific for different families of proteases were used at a concentration of 1 mM AEBSF, 100 μM antipain, 220 μM bestatin, 100 μM chymostatin, 100 μM E-64, or 10 mM EDTA.

### Immunofluorescence microscopy

Indirect immunofluorescence assays were performed as previously described [43] with anti-HA (1:2,000 dilution) or anti-histone H3 C-terminal (1:2,000 dilution) antibodies. DAPI was used to stain DNA, and images were acquired using a Olympus Fluoview FV300 microscope, software v4.3.

### Co-immunoprecipitation assays

To improve solubility for co-immunoprecipitation assays, the mononucleosome preparation from WT or WT + PfH3p-HA parasites was supplemented with 0.05% Nonidet P-40 (NP-40) and 200 mM NaCl followed by an additional extraction using RIPA buffer (Sigma R0278). Prior to immunoprecipitation, mononucleosomes were precleared by incubation with protein-G agarose beads for 1–3 h at 4°C. For immunoprecipitation, 200 μl of precleared mononucleosomes was incubated with 0.5 μg anti-HA antibodies (Roche) overnight at 4°C with rotation. Subsequently, 50 μl of protein-G agarose beads was added to each immunoprecipitation reaction and incubated for 3 h at 4°C with rotation. Bound material was washed thrice for 5 min with equilibration buffer (50 mM Tris pH 8.0, 150 mM NaCl, 5 mM EDTA pH 8.0, 0.05% NP-40) supplemented with protease inhibitors and eluted by boiling for 5 min in SDS–PAGE Laemmli buffer (37.9 mM Tris pH 6.8, 13.15% glycerol, 0.05% SDS, 0.005% bromophenol blue, and 355 mM β-mercaptoethanol). Immunoprecipitated proteins were analyzed using immunoblotting with anti-HA and anti-histone H4 antibodies.

### Chromatin immunoprecipitation and next-generation sequencing

Chromatin immunoprecipitation (ChIP) assays were performed as described previously [44] with anti-HA (Roche, 12CA5), anti-histone H3 N-terminal (H9298 Sigma), or mouse IgG (Sigma I5381) antibodies. Immunoprecipitated DNA and DNA corresponding to the ChIP input were using the Illumina Library Preparation Kit (Diagenode)

according to the manufacturer's instructions. 4–12 libraries were multiplexed and run on an Illumina HiSeq 2500 or a NextSeq 500 as a single-end run of 100 nt. The resulting fastq files were analyzed as described in the "Analysis of NGS data" section. A minimum of two biological replicates was evaluated for each ChIP assay.

### Analysis of NGS data

The analysis of ChIP-seq datasets was performed using several command line tools: BWA-MEM [45], SAMtools [46], deepTools v2.4.0, bedtools v2.26.0 [47], and MACS2 [25]. First, quality control of fastq files was performed using the FastQC software (https://www.bioinformatics.babraham.ac.uk/projects/fastqc/). Next, sequencing reads were mapped to the *P. falciparum* 3D7 genome (v3, GeneDB) using BWA-MEM under default settings to generate SAM files, which were further processed to the bam format using SAMtools. Before running MACS, the correlation between biological replicates was determined using the Pearson correlation and principal component analysis functionalities of deepTools and the normalized coverage of reads for each bam file calculated using bedtools. Thereafter, quality-filtered bam files (Q10 or Q20) served as the starting point for differential peak-calling analysis using the call peak function in MACS2. Fold enrichment of ChIP signal of treatment relative to the control was determined using the bdgcmp function in MACS2. The resulting bedgraphs were converted to bigWigs using bedGraphToBigWig [48] and the correlation between bigwig files analyzed using deepTools. Note that the mapping properties of the fastq and bam files generated in this study and related statistics are provided in Dataset EV4.

## Data availability

The fastq files supporting the results of this article are available in the EMBL-EBI European Nucleotide Archive (ENA: PRJEB18114; Sample group: ERG011046): http://www.ebi.ac.uk/ena/data/view/PRJEB18114.

**Expanded View** for this article is available online.

### Acknowledgements

This work was supported by the Consejo Nacional de Ciencia y Tecnología, Mexico (177852), the French-Mexican collaborative program PARACTIN (ANR-CONACyT Paractin 140364), and the FOSISS-CONACyT (272364) awarded to R.H.R. It was also supported by a European Research Council Advanced Grant (PlasmoSilencing 670301) and the French Parasitology consortium ParaFrap (ANR-11-LABX0024) awarded to A.S. A.M.H.S. was supported by the Consejo Nacional de Ciencia y Tecnología grant 228896, and S.S.V. was supported by the Carnot-Pasteur-Maladies Infectieuse fellowship.

### Author contributions

AS and RH-R conceived and designed the experiments; AMH-S performed mass spectrometry experiments, immunoblot, and ChIP experiments; MV developed the *in vitro* proteolysis assays; SSV and RMM performed the ChIP-seq experiments; SSV and CRM analyzed the ChIP-seq data; DL-A performed immunofluorescence assays; BX-C, GRM, and PC obtained the recombinant proteins; and SSV, AS, and RH-R wrote the manuscript. All authors read and approved the final manuscript.

## Conflict of interest

The authors declare that they have no conflict of interest.

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
