## [Review Process File · EMBO Reports]

Clipped histone H3 is integrated into nucleosomes of DNA replication genes in the human malaria parasite *Plasmodium falciparum*

Abril Marcela Herrera-Solorio, Shruthi Sridhar Vembar, Cameron Ross MacPherson, Daniela Lozano-Amado, Gabriela Romero Meza, Beatriz Xoconostle-Cazares, Rafael Miyazawa Martins, Patty Chen, Miguel Vargas, Artur Scherf and Rosaura Hernández-Rivas

Review timeline:

Submission date:	25 April 2018
Editorial Decision:	29 May 2018
Revision received:	27 August 2018
Editorial Decision:	25 September 2018
Revision received:	15 October 2018
Editorial Decision:	11 December 2018
Revision received:	21 December 2018
Accepted:	31 January 2019

Editor: Esther Schnapp

Transaction Report:

1st Editorial Decision

29 May 2018

Thank you for the submission of your research manuscript to EMBO reports. We have now received reports from the three referees that were asked to evaluate your study, which can be found at the end of this email.

As you will see, all referees think the manuscript is of interest, but referees #2 and #3 indicate that the study requires major revisions to allow publication in EMBO reports. Both referees have a number of concerns and/or suggestions to improve the manuscript, which we ask you to address in a revised manuscript. As the reports are below, I will not detail them here. We agree with referee #1 that it will be beyond the scope of this manuscript to identify the clipping protease. However, we feel that those points asking for more molecular insight should be addressed experimentally (e.g. as suggested by referee #3 in her/his point 6). Also stronger evidence for a cell-cycle-dependent cleavage of H3 should be provided (see referee #3, point 7). Finally, the technical concerns of referee #3 need to be addressed, in regarding the antibody employed for ChIP (point 5).

Given the constructive referee comments, we would like to invite you to revise your manuscript with the understanding that all referee concerns must be addressed in the revised manuscript and/or in a detailed point-by-point response. Acceptance of your manuscript will depend on a positive outcome of a second round of review. It is EMBO reports policy to allow a single round of revision only and acceptance or rejection of the manuscript will therefore depend on the completeness of your responses included in the next, final version of the manuscript.

Revised manuscripts should be submitted within three months of a request for revision; they will otherwise be treated as new submissions. Please contact us if a 3-months time frame is not sufficient for the revisions so that we can discuss the revisions further.

Supplementary/additional data: The Expanded View format, which will be displayed in the main HTML of the paper in a collapsible format, has replaced the Supplementary information. You can

submit up to 5 images as Expanded View. Please follow the nomenclature Figure EV1, Figure EV2 etc. The figure legend for these should be included in the main manuscript document file in a section called Expanded View Figure Legends after the main Figure Legends section. Additional Supplementary material should be supplied as a single pdf labeled Appendix. The Appendix includes a table of content on the first page, all figures and their legends. Please follow the nomenclature Appendix Figure Sx throughout the text and also label the figures according to this nomenclature.

For more details please refer to our guide to authors:
<http://embor.embopress.org/authorguide#manuscriptpreparation>

Important: All materials and methods should be included in the main manuscript file.

See also our guide for figure preparation:
http://www.embopress.org/sites/default/files/EMBOPress_Figure_Guidelines_061115.pdf

Please format the references according to EMBO reports style. See:
<http://embor.embopress.org/authorguide#referencesformat>

Regarding data quantification and statistics, can you please specify, where applicable, the number "n" for how many independent experiments (biological replicates) were performed, the bars and error bars (e.g. SEM, SD) and the test used to calculate p-values in the respective figure legends. Please provide statistical testing where applicable. See:
<http://embor.embopress.org/authorguide#statisticalanalysis>

We now strongly encourage the publication of original source data with the aim of making primary data more accessible and transparent to the reader. The source data will be published in a separate source data file online along with the accepted manuscript and will be linked to the relevant figure. If you would like to use this opportunity, please submit the source data (for example scans of entire gels or blots, data points of graphs in an excel sheet, additional images, etc.) of your key experiments together with the revised manuscript. Please include size markers for scans of entire gels, label the scans with figure and panel number, and send one PDF file per figure.

- a complete author checklist, which you can download from our author guidelines (<http://embor.embopress.org/authorguide#revision>). Please insert page numbers in the checklist to indicate where the requested information can be found.
- a letter detailing your responses to the referee comments in Word format (.doc)
- a Microsoft Word file (.doc) of the revised manuscript text
- editable TIFF or EPS-formatted single figure files in high resolution (for main figures and EV figures)

Please also note that we now mandate that all corresponding authors list an ORCID digital identifier that is linked to their EMBO reports account!

I look forward to seeing a revised version of your manuscript when it is ready. Please let me know if you have questions or comments regarding the revision.

REFeree REPORTS

Referee #1:

The first evidence of the epigenetic potential of histone clipping in mouse and yeast was brought 10 years ago (Duncan EM et al., Cell 2008 Santos-Rosa H et al Nat Struct Mol, 2009). While this PTM can sweep away en masse all other modifications, it was less documented likely due to the the difficulty of studying proteolysis in a well-defined biological context. Many reports only briefly mentioned detecting histone fragments, but never investigated their origin, nor their cellular

function so far. Unlike histone degradation, histone clipping as the category comprising very specific enzymatic cleavage events that have been regarded to be of potential epigenetic importance. Clipping was shown to influence chromatin compaction and might precede histone eviction. In this study, using a combination of approaches (mass spectrometry, ChIP-seq), the authors have identified clipping of the N-terminal tail of nucleosome-associated histone H3 at the position 21 as a novel PTM in *Plasmodium falciparum*. ChIP-seq analysis revealed the location of this H3 clipping in the upstream region of 6 genes involved in DNA replication and repair. While the authors identified key properties related to the inhibition of the process clipping, they failed to identify the candidate clipping protease, yet this reviewer agrees that this is beyond the scope of this manuscript. Overall, the work has been carefully performed and the results are convincing in their broad conclusion. The manuscript represents a very important amount of work, especially when one considers the difficulties to study the proteolysis of histone tails in a very complicated parasitic model, *Plasmodium falciparum*.

Referee #2:

It is an interesting report containing some novel discoveries. However, there are several questions need to be addressed for acceptance.

Major:

- 1) There is no enzymatic activity of Cathepsin C-like candidate from the parasite. So the enzyme candidate is not characterized.
- 2) No biochemistry data (binding data) to suggest Cathepsin C-like candidate against tail of H3.
- 3) No satisfying explanation on how clipping of H3 alone could regulate the related genes.
- 4) It is not very convincing why the clipping only exists within so few candidates.

Minors

Page 4, introduction section:

- 1) There is one report challenging the results of (Duncan et al., 2008), please cited it.
- 2) The authors missed several recent paradigm shifting reports regarding the endopeptidase and amino peptidase activities of JMJD5 and JMJD7.

Referee #3:

In this manuscript, Herrera Solorio and colleagues suggest that cleavage of histone 3 is a new modification of histones that possibly reflects a specific chromatin state for a small number of genes linked to replication in *Plasmodium falciparum*. This observation would be novel, but there are several points in this study that require clarification. In particular, a number of their conclusions seem to depend on an N-terminal H3 Ab whose specificity and usefulness for ChIP is not established.

1. *Plasmodium falciparum* has both an H3 and an H3.3. As in other species, the amino acid sequences of these histones are VERY similar and both N and C terminal "H3" antibodies should recognize both histones. The functions of these histones may or may not be similar in parasites, but in other eukaryotes the function of H3.3 is distinct from H3. Do they know where H3 and H3.3 are? They never address that most "H3" reagents also should recognize H3.3.
2. Figure 1 The data would be much stronger if they were able to show the results with Westerns using both the H3 N and C terminus antibodies in the whole histone preps in Figure 1 and in Figure 3 IP-Western experiments. These would more clearly establish that a cell cycle dependent H3 truncation occurs and that the reagents perform as expected.
3. Were PTM Westerns shown in 1C also performed in Ring and Troph stages?
4. They only cite one reference about the new and previously reported histone modifications in *Plasmodium falciparum* (Trelle from 2009) but more recent papers have reported other Pf histone PTM (Cobbold et al; Saraf et al) so their findings in the supplemental table S2 should be compared with these papers.

5. Since the N terminus of H3 is often heavily modified, it is not clear if this antibody actually recognizes all forms of H3 in Pf or whether it is useful for ChIP. The Sigma site is not particularly helpful and indicates that this is a useful for immunoblotting but there is no mention of this being a ChIP grade antibody. If this is not a very good ChIP Ab, one would expect that the signal would look like input and could result in data as shown in S2. Since this is ChIP-seq, it would be more informative to see the correlation coefficients of the actual data rather than the ratios.

6. Figure 2 The evidence that a Cathepsin C activity is involved is weak as you can't reliably assign a protease based on studies with a small number of general inhibitors. While identifying the protease responsible would take considerable effort, they could test whether cleavage of recombinant H3 by nuclear extract occurs if the site of H3 cleavage is mutated.

7. A pulse chase experiment would more convincingly show that there is stage specific proteolytic cleavage of H3. Although the hypothesis is plausible, it is also possible that for some reason H3 is more susceptible to cleavage during sample isolation than other histones. Further IP of soluble vs chromatin associated H3 might address whether the histone is clipped after incorporation into chromatin.

8. Doesn't their data showing agreement of ring ChIP-seq and schizont ChIP-seq of the PfH3p-HA truncated H3 conflict with the idea of H3 proteolysis being important at a particular stage of the cell cycle to coordinate karyokinesis? Maybe the truncated histone can be incorporated into a nucleosome, but does this mean that the proteolysis in the parasite occurs outside the nucleosome?

Other points:

1. It would be much easier for the reader if they used consistent symbols and color coding for ChIP replicates in PCA plots.

2. Their Westerns seem to suggest that the cleaved version of H3 has H3K27me3, a substoichiometric repressive mark in other organisms that is linked with heterochromatin. Do they think there is a link to heterochromatin as suggested in cited references?

1st Revision - authors' response

27 August 2018

Rebuttal letter to reviewers' comments (our comments are in blue-colored text):

Referee #1 :

Reply : We appreciate the very positive comments of Reviewer 1.

Referee #2:

It is an interesting report containing some novel discoveries. However, there are several questions need to be addressed for acceptance.

Major:

1) There is no enzymatic activity of Cathepsin C-like candidate from the parasite. So the enzyme candidate is not characterized.

2) No biochemistry data (binding data) to suggest Cathepsin C-like candidate against tail of H3.

Reply : The identification of the specific clipping enzyme was not the goal of this work and is beyond the scope in our opinion. Although we have evidence that a Cathepsin C like protease may be implicated in Histone H3 clipping, we believe that this is an indirect effect, since the only Cathepsin C-like nuclear protease predicted in *P. falciparum* PfDPAP2 has exopeptidase activity. PfDPAP2 may activate a so far unknown endopeptidase that cleaves the H3 N-terminal region. For this reason we feel that the suggested binding experiment will not advance the insight into the clipping molecular mechanism. All our attempts to produce a recombinant proteolytically active PfDPAP2 failed so far.

3) No satisfying explanation on how clipping of H3 alone could regulate the related genes.

Reply : We agree that our work raises many new questions linked to epigenetic regulation in malaria parasites. Several studies from different organisms suggest that histone clipping results in different phenotypes in different circumstances. However, the precise role of histone clipping is yet to be determined. We do not have a clear answer yet, but we and others observe that non-coding regulatory RNA (ncRNA) is produced from so-called intergenic regions in *P. falciparum*. H3 cleavage may interfere with ncRNA transcription. We will explore this possibility in the future. On the other hand, we observe a new histone mark H3K27me3 upstream of the cleavage site (see Fig. 1C). We will investigate in the future if H3K27me3 has a particular role on the transcription of genes that are adjacent to the observed chromatin islands of clipped H3. However, this represents a project on its own.

4) It is not very convincing why the clipping only exists within so few candidates.

Reply : We do agree with this reviewer that this is a puzzling observation. We will investigate in the future if the observed pattern is a dynamic process during sexual blood stage development and during other life cycle stages.

Minors

Page 4, introduction section:

1) There is one report challenging the results of (Duncan et al., 2008), please cited it.

Reply : We thank this reviewer for this comment. Actually, we had cited this paper (Duncan et al Cell, 2008) in the Discussion section. We added some sentences to the Discussion section to highlight the relevance of this paper for our work. (see lines 268-272)

2) The authors missed several recent paradigm shifting reports regarding the endopeptidase and amino peptidase activities of JMJD5 and JMJD7.

Reply : We thank again this reviewer for pointing out those references. We have already cited those references in the discussion section – Lines 265-266.

Referee #3:

In this manuscript, Herrera Solorio and colleagues suggest that cleavage of histone 3 is a new modification of histones that possibly reflects a specific chromatin state for a small number of genes linked to replication in *Plasmodium falciparum*. This observation would be novel, but there are several points in this study that require clarification. In particular, a number of their conclusions seem to depend on an N-terminal H3 Ab whose specificity and usefulness for ChIP is not established.

1. *Plasmodium falciparum* has both an H3 and an H3.3. As in other species, the amino acid sequences of these histones are VERY similar and both N and C terminal "H3" antibodies should recognize both histones. The functions of these histones may or may not be similar in parasites, but in other eukaryotes the function of H3.3 is distinct from H3. Do they know where H3 and H3.3 are? They never address that most "H3" reagents also should recognize H3.3.

Reply : We thank the reviewer for raising this important point. Actually, our experimental evidence supports that the clipping occurs in histone H3 and not the H3.3 variant. The peptides identified in our mass spectrometry results of the 17 kDa (histone H3 complete) and 14 kDa (Histone H3 processed) bands correspond to histone H3 and not H3.3. Some examples from Table S1 :

KSAPISAGIKK for PfH3 instead of **KSAPVSTGIK** for H3.3 and

REIAQDYKTDLRF for H3 instead of **EIAQDEYKTDLRF**

Table S1. Peptides identified on *P. falciparum* histone H3 complete and processed. The bold peptides were identified in both samples

Identified peptide	Position of aminoacids
TKQTAR	3 to 8
KSTAGKAPR	9 to 17
KSTAGKAPRK	9 to 18
KSTAGKAPRKQLASK	9 to 23
STAGKAPR	10 to 17
STGGKAPRK	10 to 18
KQLASKAAR	18 to 26
QLASKAAR	19 to 26
KSAPISAGIK	27 to 36
SAPISAGIKKPHR	28 to 40
RYQKSTDLLIR	53 to 63
STDLLIRKLPFQR	57 to 69
EIAQDYK	73 to 79
EIAQDYKTDLR	73 to 83
VTIMPKDIQLAR	117 to 128
VTIMPKDIQLAR	118 to 128

We add a sentence on Line 126 addressing this.

2. Figure 1 The data would be much stronger if they were able to show the results with Westerns using both the H3 N and C terminus antibodies in the whole histone preps in Figure 1.

Reply : We have done this experiment using mononucleosomes and it confirms our initial result. These data are now included as a new supplementary figure, Figure S1 and referred to on Lines 118-119. The numbering of the remaining supplementary figures has been modified to reflect this addition.

and in Figure 3 IP-Western experiments.

Reply : In the case of Figure 3, this experiment would not provide any additional information because the immunoprecipitation assays were done to demonstrate that the ectopically expressed clipped H3, PfH3p-HA, is integrated into nuclear mononucleosomes and also that it interacts with the histone H4.

3. Were PTM Westerns shown in 1C also performed in Ring and Troph stages?

Reply : In Figure 1A, we show that the processed form of histone H3 is generated in the schizont stage and not in Ring/Trophozoites. Since this study focuses on the processed form of histone H3, all of our subsequent assays were mainly performed on the schizont stage.

4. They only cite one reference about the new and previously reported histone modifications in *Plasmodium falciparum* (Trelle from 2009) but more recent papers have reported other PF histone PTM (Cobbold et al; Saraf et al) so their findings in the supplemental table S2 should be compared with these papers.

Reply : We thank this reviewer for pointing out these references. We now cite them together with the Trelle et al reference (see line 70). Based on the reviewer's suggestion, such a comparison was performed and histone modifications that are novel to the current study are shown in the revised version of Suppl. Table 2 (shown below and included in the modified version of the Supplementary Material document:

H3K14me2 and H3K14me3
H3K18me2,
H3K23me
H3K27me2 and H3K27me3
H3R40me
H3K56me
H3K122ac/H3K122me2

Supplementary Table S2. Summary of all Post-Translational modifications (PTMs) identified on *P. falciparum* Histone H3.

****The PTMs not identified previously are indicated in bold.**

Identify Peptide	Aminoacid	Post-traslational Modifications	PTM residues in Histone H3
KSTAGKAPR	9 to 17	Acetylation/Methylation	K9ac, K14me2
KSTAGKAPR	9 to 17	Acetylation/Methylation	K9ac, K14me3
KSTAGKAPR	9 to 17	Acetylation/Methylation	K9ac, K14me2 , R17me
KSTAGKAPR	9 to 17	Acetylation/Methylation	K9ac, K14me2 , R17me2
KSTAGKAPR	9 to 17	Acetylation/Methylation	K9ac, K14ac, K18me
KSTAGKAPR	9 to 17	Acetylation/Methylation	K9ac, K14me2 , K18me
KSTAGKAPR	9 to 17	Acetylation/Methylation	K9ac, K14me2 , K18me2
KSAPISAGIK	27 to 36	Methylation	K27me2
KSAPISAGIK	27 to 36	Methylation	K27me3
SAPISAGIKKPHR	28 to 40	Methylation	R40me
KFQKSTDLIR	53 to 63	Methylation	K56me
VTIMPKDIQLAR	117 to 128	Methylation	K122me2

5. Since the N terminus of H3 is often heavily modified, it is not clear if this antibody actually recognizes all forms of H3 in Pf or whether it is useful for ChIP. The Sigma site is not particularly helpful and indicates that this is a useful for immunoblotting but there is no mention of this being a ChIP grade antibody. If this is not a very good ChIP Ab, one would expect that the signal would look like input and could result in data as shown in S2.

Reply : We understand the reviewer's concern with the antibody used here. For this reason we have repeated the ChIP-seq experiments with ChIP-grade antibody from EMD-Millipore (06-755) on schizont stages. We find that the Millipore and Sigma anti-N-terminal Histone H3 antibodies are highly similar to each other and more similar to the input profile, and different from the anti-HA profiles (Figure 1 for this reviewer shown below). We therefore do not replace the data in the paper with these new data. We provide the Pearson correlations between the two antibodies as a figure for the reviewer (see Figure 1 below) and provide the bedgraphs for the various samples as a zip file – these files can be visualized in a genome browser such as IGV.

Figure 1 legend (for reviewer) : Correlation of ChIP-seq bam files for schizont stages of *P. falciparum* was determined using a Pearson correlation analysis in deepTools. The color scale indicates values of the Pearson correlation coefficient R from 0 to 1. Rep = Biological Replicate. These data indicate that the two antibodies against the N-terminal region of H3 behave similarly in ChIP-seq experiments.

Since this is ChIP-seq, it would be more informative to see the correlation coefficients of the actual data rather than the ratios.

Reply : To address this, we now include the correlation coefficients of the bam files in Fig. S3A and Fig. S5A.

6. Figure 2 The evidence that a Cathepsin C activity is involved is weak as you can't reliably assign a p(Ptease activity) based on studies with a small number of general inhibitors. While identifying the protease responsible would take considerable effort, they could test whether cleavage of recombinant H3 by nuclear extract occurs if the site of H3 cleavage is mutated.

Reply : As mentioned earlier in reply to reviewer 2, point 2, we do not believe that the Cathepsin C-like activity defines the clipping protease. We hypothesize that the predicted plasmodial Cathepsin C is encoded by the exopeptidase PfDPAP2 (according to PlasmoDB) and PfDPAP2 may activate an endopeptidase that provides the observed histone clipping activity. Regardless, we have considered this request and have produced two mutants of histone H3 at the cleavage site, i.e., amino acid 21-22 : we replaced Alanine21/Serine22 by Isoleucine21/Threonine22 or Valine21/Threonine22. Wild-type or mutant GST-H3 fusion proteins were incubated with schizont-stage nuclear extracts for 0, 15, 30 and 90 minutes. Western blot analysis with anti-GST antibodies were used to measure cleavage efficacy. We observed cleavage of the mutant proteins in a manner similar to wildtype (Figure 2 see below). These results are shown as figure 2 for this reviewer. Since we have not yet identified the endoprotease that cleaves H3, we can only speculate about the molecular events that lead to clipping. For example, mouse H3 clipping was reported to initiate between position 21 and 22 of the amino terminus, while the final cleavage occurs between amino acids 27 and 28 (Duncan et al., 2008). Based on this results, we speculate that a combination of endo and exopeptidase may be involved in the observed cleavage in *Plasmodium* H3. The endopeptidase could cleave somewhere before the Ala21 and then trimming by an exopeptidase may occur. This hypothesis could explain that the mutants are cleaved in a similar way to the wildtype GST-fusion protein. Actually, we observe an intermediate H3 cleaved band that could represent an initial endoprotease-mediated clipping event of plasmodial H3 (see Figure 1 B and C in our manuscript),

before a secondary proteolytic process (potentially by an exopeptidase) occurs leading to the final H3 truncated protein. Clearly, more in depth analysis is needed to explore the underlying molecular events of clipping.

If this reviewer thinks that those data should be part of the revised manuscript, we are happy to add this into the supplementary data section. However, we feel that much more effort is needed to identify the protease(s) that contribute to the N-terminal clipping we describe here.

Figure 2. : A) Cleavage site of histone H3 identified in this study and mutations introduced at the cleavage site. B) 1 µg of each recombinant histone representing wild type H3, two mutant H3 and H2A were mixed with 20 µg of nuclear extract from late stage 3D7 parasites. Samples were incubated at 37°C for different times (0, 15, 30 and 90 minutes). The samples were ran on TGX Stain-Free gels and imaged with ChemiDoc XRS molecular imager (BioRad). Western Blots were performed with anti-GST antibody to reveal recombinant histone species at 40kDa.

7. A pulse chase experiment would more convincingly show that there is stage specific proteolytic cleavage of H3. Although the hypothesis is plausible, it is also possible that for some reason H3 is more susceptible to cleavage during sample isolation than other histones.

Reply : Our nuclear extracts used in the clipping experiments are prepared freshly in the presence of a cocktail of protease inhibitors. We do not think that there is any rationale to support that histone H3 in later stages is more susceptible to cleavage, neither is there any support for this in the literature.

Further IP of soluble vs chromatin associated H3 might address whether the histone is clipped after incorporation into chromatin.

Reply : This is still an open question in the literature. In the histone clipping field it is speculated that both scenarios do exist (see recent review : Azad and Tomar, Mol Biol Rep 2014). This manuscript provides to our knowledge the first experimental evidence that free clipped histone H3 (expressed from an episome) can be incorporated into nucleosomes. Apparently histones are rapidly incorporated into nucleosomes. Thus, it is far from evident that the suggested experiments will address this question without developing new challenging protocols for *Plasmodium*. We feel that this is out of scope for this manuscript.

8. Doesn't their data showing agreement of ring ChIP-seq and schizont ChIP-seq of the PfH3p-HA truncated H3 conflict with the idea of H3 proteolysis being important at a particular stage of the cell cycle to coordinate karyokinesis? Maybe the truncated histone can be incorporated into a nucleosome, but does this mean that the proteolysis in the parasite occurs outside the nucleosome?

Reply : We agree with the reviewer's observation. When we express the truncated H3 during the entire asexual blood stage cycle, the cellular machinery inserts clipped H3 into the chromatin already at ring stage development. However, the episomal clipped H3 expression profile does not correspond to the observed clipping profile of endogenous H3. It shows that the cellular machinery that deposits truncated H3 into nucleosomes is active at ring stage. We do not exclude that both pathways exist in malaria parasites: i) proteolytically processed free H3 and ii) proteolytically processed nucleosomal H3. To explore this question will need much more in depth studies which is out of scope for a study that describes for the first time clipping in malaria parasites.

Other points:

1. It would be much easier for the reader if they used consistent symbols and color coding for ChIP replicates in PCA plots.

Done

2. Their Westerns seem to suggest that the cleaved version of H3 has H3K27me3, a substoichiometric repressive mark in other organisms that is linked with heterochromatin. Do they think there is a link to heterochromatin as suggested in cited references?

Reply : We thank the reviewer for pointing out the potential role of H3K27me3 in processing of H3. Effectively, our data (Figure 1C) suggest that the processed form of histone H3 has a putative repressive mark at lysine 27 (H3K27me3). More work is needed to explore the role of H3K27me3 either in the clipping or deposition process into nucleosomes.

In *P. falciparum*, heterochromatin has been linked to H3K9me3 and H3K36me marks. The role of H3K27me3 has not been explored yet. Thus, it remains to be seen if this mark is associated to heterochromatin islands in malaria parasites. ChIP-seq analysis using antibodies against this mark could provide interesting insight into clipping and heterochromatin. This is an interesting experiment we will follow up in the future.

2nd Editorial Decision

25 September 2018

Thank you for the submission of your revised manuscript. We have now received the enclosed comments from the referee who was asked to assess it. I am happy to tell you that we can in principle accept your manuscript now.

A few more minor changes are still needed:

Your manuscript has 4 figures and we will therefore publish it as a short report. Short reports have combined results and discussion sections, this needs to be corrected.

Please provide up to 5 keywords.

Please change the reference style to the numbered EMBO reports style that can be found in EndNote.

Please upload all figures as separate files.

The manuscript text does not call out figure 4C.

Table 1 has colored text, but we can only process white, black and grey colors for tables in the main manuscript file.

The Appendix needs to be uploaded as a single file with a table of content, and the figures and tables

need to be called Appendix Figure S1, Appendix Table S1, etc.

The supplementary tables can either be EV tables (Table EV1, etc) or Datasets (Dataset EV1, etc). If the tables are source data for any of the figures, they need to be called "Source data for figure X". Table S6 should rather be a Dataset.

EMBO press papers are accompanied online by A) a short (1-2 sentences) summary of the findings and their significance, B) 2-3 bullet points highlighting key results and C) a synopsis image that is 550x200-400 pixels large (the height is variable). You can either show a model or key data in the synopsis image. Please note that text needs to be readable at the final size. Please send us this information along with the revised manuscript.

I would like to suggest a few minor changes to the abstract that needs to be written in present tense. Please let me know whether you agree with the following:

Post-translational modifications of histone H3 N-terminal tails are key epigenetic regulators of virulence gene expression and sexual commitment in the human malaria parasite *Plasmodium falciparum*. Here, we identify proteolytic clipping of the N-terminal tail of nucleosome-associated histone H3 at amino acid position 21 as a new chromatin modification. A Cathepsin C-like proteolytic clipping activity is observed in nuclear parasite extracts. Notably, an ectopically expressed version of clipped histone H3, PfH3p-HA, is targeted to the nucleus and integrates into mononucleosomes. Furthermore, chromatin immunoprecipitation and next generation sequencing analyses show that PfH3p-HA is highly enriched in the upstream region of six genes that play a key role in DNA replication and repair: in these genes, PfH3p-HA demarcates a specific 1.5 kb chromatin island adjacent to the open reading frame. Our results indicate that, in *P. falciparum*, the process of histone clipping may precede chromatin integration hinting at preferential targeting of pre-assembled PfH3p-containing nucleosomes to specific genomic regions. The discovery of a protease-directed mode of chromatin organization in *P. falciparum* opens up new avenues to develop new anti-malarials.

I look forward to seeing a final revised version of your manuscript as soon as possible.

REFEREE REPORT

Referee #3:

The authors have addressed my prior comments.

2nd Revision - authors' response

15 October 2018

We have finalized the revision as suggested by you. All points raised in your last email have been incorporated into the submitted manuscript

3rd Editorial Decision

11 December 2018

I am glad that the image aberrations in your manuscript have been clarified. We would like to proceed with the publication of your study, however, a few more changes are needed.

Please send us all gel figures and source data at a higher resolution. In order to do so, you might have to re-scan the gels at higher resolution. Please also do not use the "smoothing blur application" when preparing the figures and please leave clear spaces between the lanes of the weight markers (e.g. Fig 2 and Fig 3).

The Appendix is a mix of different files now and this needs to be clarified:

- Please remove all info on EV tables and Datasets from the Appendix
- EV tables should start with and be called Table EV1, Table EV2, etc and need to have the title and legend within the excel file of the table
- The Dataset needs to be called Dataset EV1 (it is not part of the Appendix)
- Appendix Figs S5 and S6 are missing in the current file, please add
- IF Appendix Tables S1 and S2 are source data for Fig1 they need to be called "Source data for Fig1" and taken out of the Appendix. The source data for one figure need to be uploaded as a single, separate pdf file that can have several pages.
- delete "Supplementary Table" from the titles in Tables EV3- EV5. The EV tables can have several tabs with different titles if necessary.
- Please update and add callouts for all EV tables in the manuscript text
- Please increase the resolution of the Appendix figures
- Please send us Table 1 in either word or excel file

4th Revision - authors' response

24 January 2019

We thank the editorial team for their patience and continual support to clarify issues with the Western blot results. We have sorted out those problems mentioned in your last email from 8/01/2019 and have submitted now the corrected figures in the requested resolution with their corresponding source files. We have also changed the reference style to Embo reports and have deleted the "data not shown section" in the indicated sections.

Corresponding Author Name: Hernández-Rivas Rosaura

Journal Submitted to: EMBO Report

Manuscript Number: EMBOR-2018-46331V2